# Assessing drug safety by identifying the axis of arrhythmia in cardiomyocyte electrophysiology

**Stewart Heitmann[1]\*, Jamie I Vandenberg[1,2], Adam P Hill[1,2,3]**

[1]Victor Chang Cardiac Research Institute, Darlinghurst, Australia; [2]School of Clinical Medicine, Faculty of Medicine and Health, University of New South Wales, Sydney, Australia; [3]Victor Chang Cardiac Research Institute Innovation Centre, Darlinghurst, Australia

**Abstract** Many classes of drugs can induce fatal cardiac arrhythmias by disrupting the electrophysiology of cardiomyocytes. Safety guidelines thus require all new drugs to be assessed for pro-arrhythmic risk prior to conducting human trials. The standard safety protocols primarily focus on drug blockade of the delayed-rectifier potassium current ($I_{Kr}$). Yet the risk is better assessed using four key ion currents ($I_{Kr}$, $I_{CaL}$, $I_{NaL}$, $I_{Ks}$). We simulated 100,000 phenotypically diverse cardiomyocytes to identify the underlying relationship between the blockade of those currents and the emergence of ectopic beats in the action potential. We call that relationship the axis of arrhythmia. It serves as a yardstick for quantifying the arrhythmogenic risk of any drug from its profile of multi-channel block alone. We tested it on 109 drugs and found that it predicted the clinical risk labels with an accuracy of 88.1–90.8%. Pharmacologists can use our method to assess the safety of novel drugs without resorting to animal testing or unwieldy computer simulations.

## eLife assessment

This **compelling** and novel mathematical method assesses drug pro-arrhythmic cardiotoxicity by examining the electrophysiology of untreated cardiac cells. It will be **valuable** for future drug safety design.

## Introduction

Torsades des Pointes is a potentially lethal ventricular arrhythmia that can be induced by many classes of drugs. These include antibiotics, antipsychotics, antihistamines, chemotherapeutics, and anti-arrhythmics (*Yap and Camm, 2003*). The majority of torsadogenic drugs block the hERG ion channel which carries the delayed-rectifier potassium current ($I_{Kr}$) (*Witchel, 2011*). For this reason, international safety guidelines require hERG block to be assessed in living cells prior to conducting human trials (*ICH, 2005*). However, the standard hERG assay is overly sensitive. It does not accommodate multi-channel effects which render some drugs safe despite blocking hERG (*Martin et al., 2004*; *Hoffmann and Warner, 2006*). Consequently, many useful drugs are prematurely abandoned during pre-clinical trials. The safety pharmacology community is actively pursuing new in vitro and in silico assays that improve accuracy by targeting multiple ion channels (*Pugsley et al., 2008*; *Colatsky et al., 2016*).

In silico assays use computational models of cardiomyocyte electrophysiology in place of a living cell (*Mirams et al., 2011*; *Lancaster and Sobie, 2016*; *Mann et al., 2016*; *Dutta et al., 2017*; *Passini et al., 2017*; *Ballouz et al., 2021*; *Llopis-Lorente et al., 2020*). Drug blockade (*Figure 1A*) is simulated

**\*For correspondence:**
s.heitmann@victorchang.edu.au

**Competing interest:** The authors declare that no competing interests exist.

**Figure 1.** Conceptual framework. (**A**) Drugs simultaneously block multiple species of ion channels to differing degrees. The principal ion currents implicated in drug-induced Torsades des Pointes are $I_{CaL}, I_{Kr}, I_{NaL}$, and $I_{Ks}$. (**B**) Simplified circuit diagram of cardiomyocyte electrophysiology. Drug blockade is simulated by attenuating the ionic conductances ($G_{CaL}, G_{Kr}, G_{NaL}, G_{Ks}$). Those parameters are also varied randomly to mimic individual differences in electrophysiology. (**C**) Simulated action potentials of phenotypically diverse cardiomyocytes. Early after-depolarizations (red) are biomarkers for Torsades des Pointes. Conventional in silico assays simulate the effect of drugs on cardiomyocytes on a case-by-case basis. Our method inverts the procedure by simulating cardiomyocytes in the absence of drugs and then inferring how drugs would behave.

in the model (*Figure 1B*) by attenuating the conductivity of the relevant ion currents. The conductance parameters are labeled $G$ by convention. The simulations can be repeated across a diverse range of cardiac phenotypes to ensure generality of the results (*Britton et al., 2013*; *Ni et al., 2018*; *Gong and Sobie, 2018*). The individual phenotypes are constructed by randomizing the conductance parameters to mimic natural variation in ion channel expression. The method produces a population of cardiac action potentials (*Figure 1C*). Selected biomarkers within the action potentials are then statistically analyzed for drug-induced changes (e.g. *Hondeghem, 2005*; *Varshneya et al., 2021*). Contemporary research is largely concerned with improving those biomarkers.

Yet the main problem with the conventional approach is that it requires multitudes of computationally intensive simulations for every drug that is assessed. Pharmacology laboratories must invest heavily in specialist computing resources and expertise before they can apply the methods to their drugs. We propose a new approach that allows drugs to be assessed without conducting drug-specific simulations. The method is initiated by simulating a diverse population of cardiomyocytes in the absence of drugs. That simulation need only be done once. The drug-free population is then used to identify the principal relationship between ionic conductances and ectopic phenotypes. We call that relationship the *axis of arrhythmia* because it describes the principal pathway for transforming benign

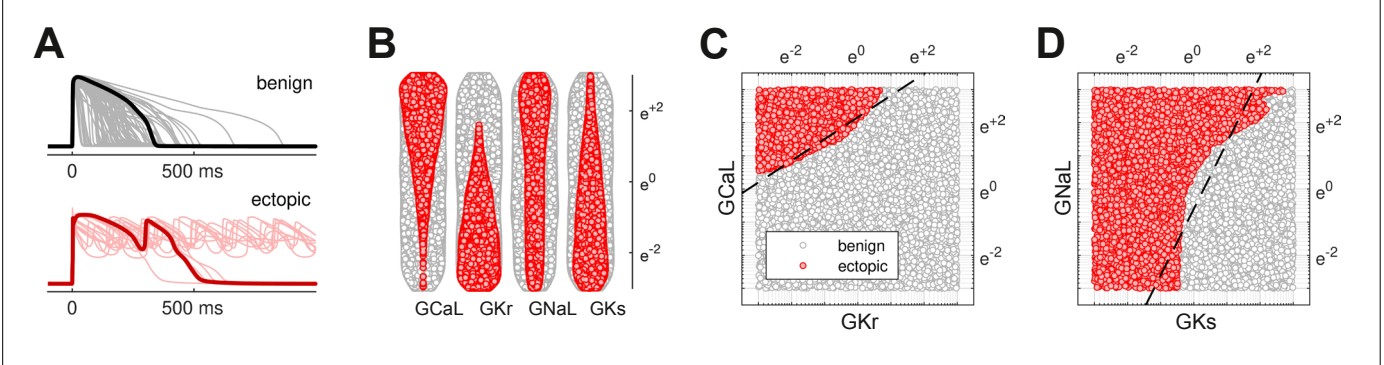

**Figure 2.** Benign versus ectopic cardiac phenotypes. (**A**) Simulated action potentials for cardiomyocytes with randomly scaled conductance parameters $G_{Kr}, G_{CaL}, G_{NaL}$, and $G_{Ks}$. Myocytes that exhibited early after-depolarizations were classified as *ectopic* (red). Those that did not were classified as *benign* (gray). (**B**) Swarm plots of the conductance scalars on a logarithmic scale. Color indicates the classification of the myocyte (benign versus ectopic). (**C**) Two-dimensional slice of parameter space showing the relationship between ectopic and benign phenotypes in $G_{CaL}$ versus $G_{Kr}$. The dashed line is the statistical decision boundary. $G_{NaL}$ and $G_{Ks}$ were fixed at unity ($e^0 = 1$). (**D**) Two-dimensional slice showing $G_{NaL}$ versus $G_{Ks}$. In this case $G_{CaL} = e^{0.46}$ and $G_{Kr} = e^{-2.3}$.

cardiomyocytes into ectopic cardiomyocytes. Thereafter, the axis serves as a yardstick for assessing the torsadogenic risk of drugs directly from their physiological signature of ion channel blockade alone.

## Results

The action potentials of 100,000 randomized ventricular cardiomyocytes (*Figure 2A*) were simulated using a variant of the O'Hara-Rudy model (*O'Hara et al., 2011*) that was optimized for long QT syndrome (*Mann et al., 2016*; *Krogh-Madsen et al., 2017*). Early after-depolarizations were chosen as a biomarker for Torsades des Pointes. Cardiomyocytes that exhibited early after-depolarizations were classified as *ectopic* (red) and those that did not were classified as *benign* (gray).

The four cardiac ion currents that we investigated ($I_{CaL}, I_{Kr}, I_{NaL}, I_{Ks}$) have previously been implicated in torsadogenic risk (*Dutta et al., 2017*; *Llopis-Lorente et al., 2020*). The myocytes were constructed by re-scaling the conductance parameters ($G_{CaL}, G_{Kr}, G_{NaL}, G_{Ks}$) with randomly selected multipliers that were drawn uniformly from a logarithmic scale (*Figure 2B*). The use of the logarithmic coordinate frame is crucial to our subsequent analysis.

The ectopic and benign phenotypes were clearly segregated in parameter space (*Figure 2C and D*). Multivariate logistic regression was used to identify the linear boundary (dashed line) that best separated the two classes. We refer to it as the *decision boundary* following the conventions of classification statistics.

### Decision boundary

The multivariate logistic regression equation,

$$\ln \frac{p}{1-p} = \beta_0 + \beta_{CaL}X_{CaL} + \beta_{Kr}X_{Kr} + \beta_{NaL}X_{NaL} + \beta_{Ks}X_{Ks}, \tag{1}$$

describes the log-odds of a cardiomyocyte being ectopic where $X_{CaL} = \ln(G_{CaL})$, $X_{Kr} = \ln(G_{Kr})$, $X_{NaL} = \ln(G_{NaL})$, and $X_{Ks} = \ln(G_{Ks})$. The decision boundary is the hyperplane in four-dimensional parameter space ($X_{CaL}, X_{Kr}, X_{NaL}, X_{Ks}$) where p = 0.5. *Figure 2C and D* shows the intersection of that hyperplane (dashed) with two-dimensional slices of parameter space. Although we illustrate the concept in two dimensions, the analysis itself is conducted in four dimensions.

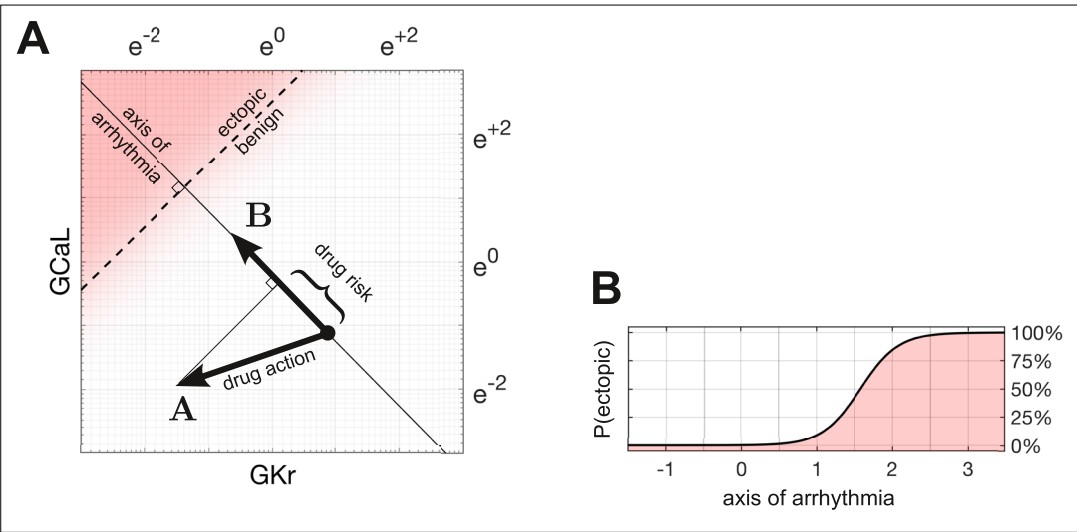

**Figure 3.** Quantifying drug risk with the axis of arrhythmia. (**A**) The axis of arrhythmia runs orthogonally to the decision boundary. As such, it describes the shortest pathway to ectopy for any cardiomyocyte. The basis vector of the axis is labeled **B**. The action of the drug is labeled **A**. The arrhythmogenic component of the drug is obtained by projecting vector **A** onto vector **B**. The length of the projection is our measure of drug risk. (**B**) The probability of ectopy along the axis of arrhythmia. The origin corresponds to the baseline cardiomyocyte. The distance from the origin corresponds to the risk score. Distance is measured in log units, using the same scale as panel A.

Our estimates of the regression coefficients, $\beta_0 = -6.416 \pm 0.059$, $\beta_{\text{CaL}} = 2.509 \pm 0.024$, $\beta_{\text{Kr}} = -2.471 \pm 0.024$, $\beta_{\text{NaL}} = 0.847 \pm 0.012$, $\beta_{\text{Ks}} = -1.724 \pm 0.018$, were all statistically significant (p < 0.001). The reported confidence intervals represent ± 1 SE. The model itself was also significantly different from the null model, $\chi^2(99995, n = 100000) = 73900$, p < 0.001.

### The action of drugs in logarithmic coordinates

Drugs attenuate the conductivity of ion channels in a multiplicative fashion. The conductance of the drugged channel is defined as

$$G_{\text{drug}} = \delta \times G_{\text{ion}},$$

where $G_{\text{ion}}$ is the baseline conductance of the ion channel and $\delta \in [0, 1]$ is the fractional conductance imposed by the drug. Logarithmic coordinates transform the action of drug blockade into an additive process,

$$\ln(G_{\text{drug}}) = \ln(\delta) + \ln(G_{\text{ion}}),$$

which can be expressed with vector geometry (*Figure 3A*). We denote the action of a drug by the vector,

$$\mathbf{A} = \{\alpha_{\text{CaL}}, \alpha_{\text{Kr}}, \alpha_{\text{NaL}}, \alpha_{\text{Ks}}\},$$

where $\alpha = \ln(\delta)$. The values of $\delta$ would ordinarily be obtained from patch clamp experiments. In our case, we calculated them from the published potencies of 109 drugs collated by *Llopis-Lorente et al., 2020*.

### Axis of arrhythmia

We define the axis of arrhythmia as a line that runs orthogonally to the decision boundary (*Figure 3A*). It represents the shortest path for shifting any cardiomyocyte into the ectopic regime by modifying the conductances of its ion channels. The basis vector of the axis,

$$\mathbf{B} = \{\beta_{\text{CaL}}, \beta_{\text{Kr}}, \beta_{\text{NaL}}, \beta_{\text{Ks}}\}, \tag{2}$$

is defined by the coefficients of the regression equation.

### Drug risk metric

The arrhythmogenic component of a drug is obtained by projecting the action of the drug onto the axis of arrhythmia. The length of the projection is our metric of drug risk. Specifically,

$$\text{risk} = \frac{\mathbf{A} \cdot \mathbf{B}}{\|\mathbf{B}\|}, \tag{3}$$

where $\mathbf{A}$ is the action of the drug, $\mathbf{B}$ is the basis vector of the axis of arrhythmia, and

$$\mathbf{A} \cdot \mathbf{B} = \sum_i \alpha_i \beta_i$$

is the dot product. The metric is normalized to the Euclidean length of $\mathbf{B}$, which is denoted $\|\mathbf{B}\|$. From our regression coefficients,

$$\mathbf{B} = \{2.509, -2.471, 0.847, -1.724\}$$

and $\|\mathbf{B}\| = \sqrt{\beta_{\text{CaL}}^2 + \beta_{\text{Kr}}^2 + \beta_{\text{NaL}}^2 + \beta_{\text{Ks}}^2} = 4.01$.

### Risk scores

According to our metric, drugs that shift the electrophysiology *toward* the ectopic region have *positive* scores, whereas drugs that shift it *away* from ectopy have *negative* scores. Drugs that do neither have scores near zero.

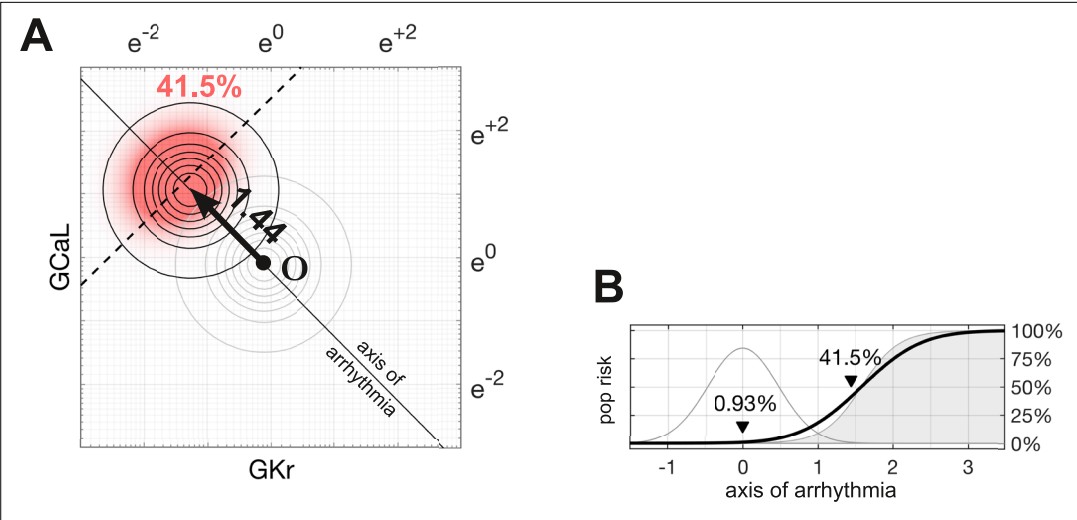

**Figure 4.** Susceptibility to a drug in the natural population. (**A**) Natural variation in ion channel conductivity is represented by a symmetric Gaussian density function centered at point **O**. In this example, a 10-fold dose of Ibutilide shifts the population by 1.44 units toward the ectopic region. The proportion of ectopic myocytes in the drugged population is 41.5% (red). (**B**) The relationship between the drug risk score and ectopy in the natural population. The drug risk score corresponds to position on the axis of arrhythmia. The shaded region is the a priori probability of ectopy along that axis (reproduced from *Figure 3B*). The Gaussian profile (thin gray line) is the natural population density centered at zero. The proportion of myocytes that are ectopic (heavy black line) is 0.93% at baseline. That proportion rises as the drug shifts the population density toward the decision boundary.

The online version of this article includes the following video for figure 4:

**Figure 4—video 1.** An animated version of this figure showing how the population density shifts as the risk score increases.

https://elifesciences.org/articles/90027/figures#fig4video1

## The a priori probability of ectopy

The probability of ectopy along the axis of arrhythmia,

$$p(x) = \frac{1}{1 + \exp(x)}, \tag{4}$$

is obtained by rearranging *Equation 1* and substituting $x = \beta_0 + \beta_{\text{CaL}}X_{\text{CaL}} + \beta_{\text{Kr}}X_{\text{Kr}} + \beta_{\text{NaL}}X_{\text{NaL}} + \beta_{\text{Ks}}X_{\text{Ks}}$. *Equation 4* describes the a priori probability of a cardiomyocyte being ectopic, based on its proximity to the decision boundary (*Figure 3B*). The shallow slope of the profile reflects the uncertainty in fitting a linear boundary to the data, as seen in *Figure 2C and D*.

## Susceptibility in the natural population

Any given drug alters the electrophysiology of all cardiomyocytes to the same extent, but only some cardiomyocytes become ectopic. The most susceptible are those that are closest to the decision boundary. We calculated the proportion of the natural population that would be susceptible to a given drug by analyzing how the drug shifts the population density with respect to the decision boundary.

Following the methods of *Sobie, 2009*, *Sadrieh et al., 2013*, *Morotti and Grandi, 2017*, *Gong and Sobie, 2018*, we assumed that ion channel conductances varied independently and were log-normally distributed ($\mu = -0.112, \sigma = 0.472$). A log-normal distribution maps onto a normal distribution in logarithmic coordinates, by definition. The natural population in our parameter space is therefore a symmetric multivariate Gaussian density function (*Figure 4A*). In the absence of drugs, the natural population density in four dimensions is centered at the point $\mathbf{O} = \{\mu, \mu, \mu, \mu\}$.

The proportion of myocytes that become ectopic depends on how far the population is shifted along the axis of arrhythmia according to the risk metric (*Figure 4B*). The proportion is calculated as the product of the a priori probability of the ectopy (*Equation 4*) and the density of the drugged population (*Equation 6*; Methods). For the case of 10× therapeutic dose of Ibutilide — which is a

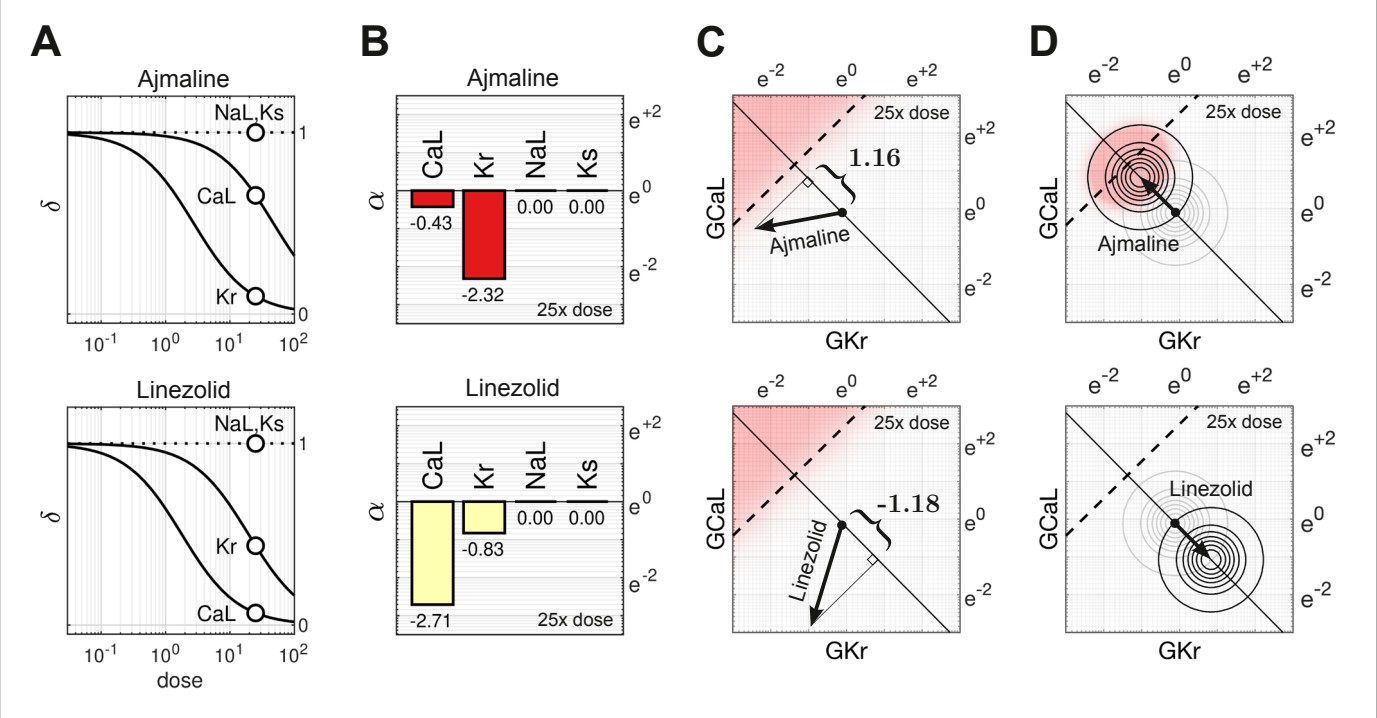

**Figure 5.** Cases of Ajmaline and Linezolid. (**A**) Drug-response profiles of $I_{CaL}$, $I_{Kr}$, $I_{NaL}$, and $I_{Ks}$ relative to the therapeutic dose. Open circles highlight 25× therapeutic dose. For Ajmaline, $\delta_{CaL} = 0.654$ and $\delta_{Kr} = 0.0986$ at 25× dose. For Linezolid, $\delta_{CaL} = 0.067$ and $\delta_{Kr} = 0.437$ at 25× dose. Data for $I_{NaL}$ and $I_{Ks}$ were not available, so those channels were assumed to be unaffected by the drugs in both cases ($\delta_{NaL} = 1$ and $\delta_{Ks} = 1$). (**B**) The blocking action of Ajmaline and Linezolid at 25× dose. By definition, $\alpha = \ln(\delta)$. (**C**) The corresponding risk scores for Ajmaline (+1.16) and Linezolid (−1.18) at that dose. (**D**) The drug-induced shifts of the natural population density along the axis of arrhythmia. The proportion of myocytes that are ectopic with 25× dose of Ajmaline is 26% (red), compared to only 0.0095% for Linezolid.

potent $I_{Kr}$ blocker — the proportion of the natural population that is susceptible to drug-induced Torsades is 41.5% (*Figure 4*). The size of the susceptible population is a monotonic function of the drug's risk score, so the torsadogenic risk can be described using either terminology.

## Validation against known drugs

We used the Hill equation to reconstruct the drug-response profiles of $G_{Kr}$, $G_{CaL}$, $G_{Ks}$, and $G_{NaL}$ from the half-maximal inhibitory concentrations ($IC_{50}$) of 109 drugs reported by *Llopis-Lorente et al., 2020*. They labeled the clinical risks according to the Credible Meds list of QT drugs (*Woosley et al., 2019*). In that labeling scheme, Class 1 drugs carry a known torsadogenic risk; Class 2 drugs carry a possible risk; Class 3 drugs carry a risk but only in conjunction with other factors; Class 4 drugs have no evidence of risk at all. The reconstructed drug-response profiles, along with their corresponding risk scores, are provided in *Table 2—source data 1* and plotted in *Table 2—source data 2*.

## Cases of Ajmaline and Linezolid

Ajmaline (*Figure 5A*, top) is an anti-arrhythmic drug that slows conduction by blocking the fast sodium current, $I_{Na}$ (*Kiesecker et al., 2004*). It also blocks $I_{Kr}$, which is the reason Ajmaline has a known risk (Class 1) of inducing Torsades. In comparison, Linezolid (*Figure 5A*, bottom) is an antibacterial agent that has no clinical evidence of Torsades (Class 4) even though it too blocks $I_{Kr}$, albeit to a lesser extent than it blocks $I_{CaL}$. Indeed, the two drugs have nearly opposite effects on $G_{CaL}$ and $G_{Kr}$, as can be seen in *Figure 5B*.

At 25× therapeutic dose, the blocking action of Ajmaline is written in vector notation as

$$\mathbf{A}_{Ajmaline} = \{-0.425, -2.32, 0, 0\}.$$

Likewise, the blocking action of Linezolid is

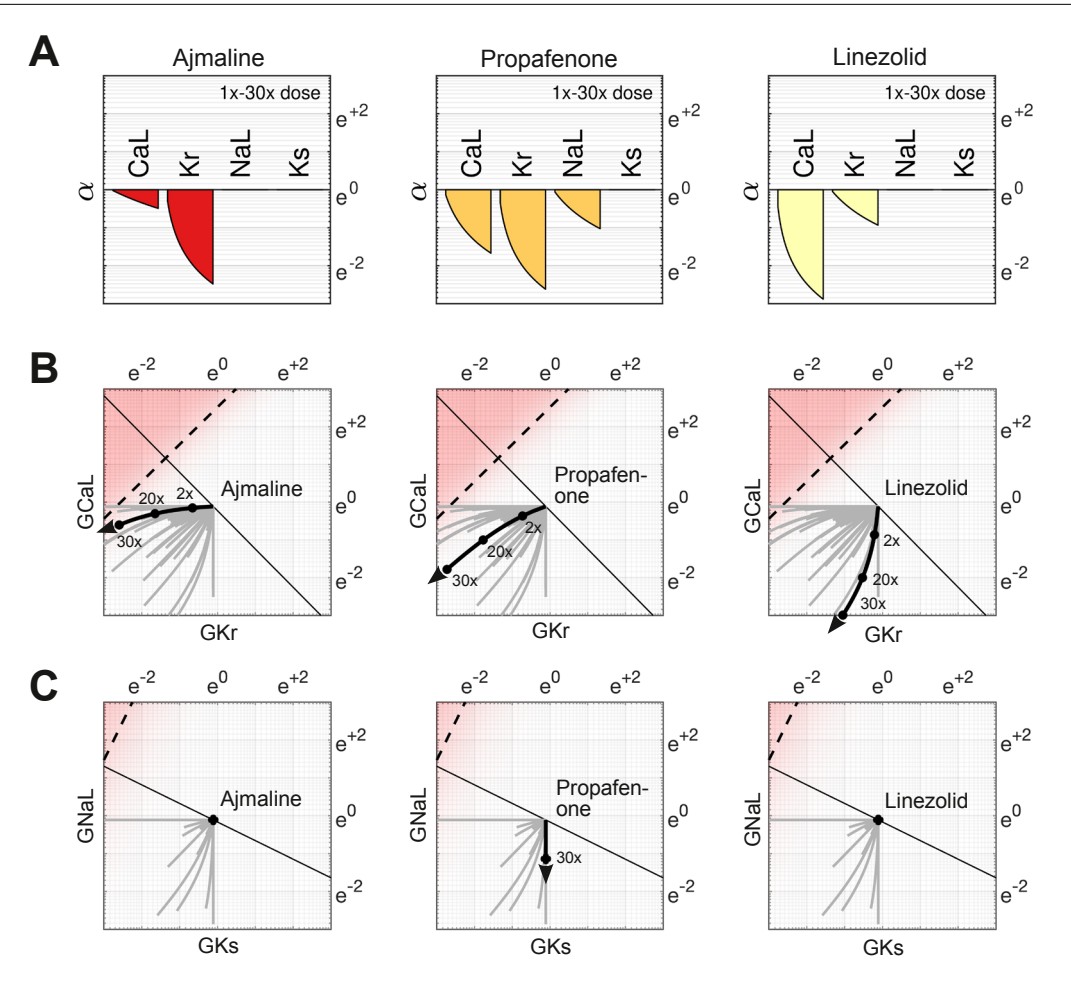

**Figure 6.** Effect of dose on multi-channel drug-block. (**A**) The attenuation of $G_{CaL}$, $G_{Kr}$, $G_{NaL}$, and $G_{Ks}$ for 1× to 30× therapeutic doses of Ajmaline, Propafenone, and Linezolid, respectively. The ion channels respond to dosage at differing rates. (**B**) The dose-dependent action of each drug in the parameter space of $G_{CaL}$ and $G_{Kr}$. The paths are curvilinear because of the differing response rates of the ion channels. For comparison, the gray traces are the pathways of all 109 drugs in the dataset. (**C**) The corresponding pathways in $G_{NaL}$ and $G_{Ks}$. Propafenone attenuates $G_{NaL}$ but not $G_{Ks}$. Neither Ajmaline nor Linezolid attenuate $G_{NaL}$ or $G_{Ks}$, but other drugs do (gray traces).

$$\mathbf{A}_{Linezolid} = \{-2.71, -0.828, 0, 0\}.$$

The two drugs shift the electrophysiology in opposite directions along the axis of arrhythmia (*Figure 5C*). The corresponding risk scores are $+1.16$ for Ajmaline and $-1.18$ for Linezolid. The signs of the scores suggest that Ajmaline is pro-arrhythmic whereas Linezolid is not, which agrees with the clinical risk labels. Indeed, the sizable negative score for Linezolid suggests that it may actually have anti-arrhythmic properties.

The effect of both drugs on the natural population is shown in *Figure 5D*. Ajmaline shifts the population density by 1.16 units toward the ectopic regime, making 26% of the population susceptible to Torsades. Conversely, Linezolid shifts the population 1.18 units away from the ectopic regime, making only 0.0095% of the population susceptible, which is a substantial drop compared to the baseline rate of 0.93%.

## The effect of dosage on multi-channel block

The action of a drug subtly changes direction with dosage because of differing response rates for each ion channel. For example, Ajmaline (*Figure 6A*, left) disproportionately blocks more $I_{Kr}$ than

$I_{CaL}$ with increasing dosage. Its dose-dependent action thus follows a gentle arc in parameter space rather than a straight line (**Figure 6B**, left). The effect is less pronounced for Propafenone (**Figure 6A**, middle) which recruits the ion channels more uniformly. Nonetheless, its path is still not strictly linear (**Figure 6B**, middle). Linezolid (**Figure 6B**, right) follows a similar pattern to Ajmaline (**Figure 6B**, left), but in the opposite direction. Of these three drugs, only Propafenone blocks $I_{NaL}$ and none block $I_{Ks}$, hence their pathways in $G_{Ks}$ and $G_{NaL}$ (**Figure 6C**) are of little consequence. Nonetheless, the pathways of other drugs (gray traces) are often curvilinear in four dimensions ($G_{CaL}$, $G_{Kr}$, $G_{NaL}$, $G_{Ks}$). That curvature influences how these drugs project onto the axis of arrhythmia, hence the risk scores depend on dosage. The shifting scores reflect the changing balance of ionic currents that occur naturally with multi-channel drugs.

## Testing the risk metric

The metric was tested by scoring all 109 drugs over a range of doses and comparing the results to the clinical risk labels from Credible Meds. The clinical labels were lumped into two categories for this purpose: UNSAFE (Classes 1 and 2) versus SAFE (Classes 3 and 4). Drugs that scored above a given threshold ($risk > \theta$) were predicted to be unsafe and those that scored below the threshold ($risk < \theta$) were predicted to be safe. The threshold was optimized for each dosage. For example, a classification accuracy of 90.8% was achieved for drugs at 25× dose using a scoring threshold of $\theta = 0.195$ (**Figure 7**).

The procedure was repeated for doses ranging from 1× to 32×. The classification accuracy for all dosage levels was found to lay between 88.1% and 90.8% (**Figure 8A**). The differences were primarily due to borderline cases, so we refrain from nominating any one dosage as being optimal. In comparison, the conventional hERG assay has an accuracy of 78.9% for the same dataset (**Llopis-Lorente et al., 2020**).

From a safety perspective, the trade-off between false negatives and false positives can be tuned by adjusting the scoring threshold, $\theta$. This is illustrated by the receiver operating characteristic (ROC) curves (**Figure 8B and C**). In our case, there is little difference between the ROC curves for drugs assessed at 5× dose versus 25× dose. The areas under the respective ROC curves (AUROC) are nearly identical at 91.3% and 91.5%. The conventional hERG assay, in comparison, has an AUROC of $77 \pm 7\%$ (**Kramer et al., 2013**).

## Discussion

In this study, we have proposed a new metric of torsadogenic risk that is based on the axis of arrhythmia. The major benefit of the metric is that it can be applied to novel drugs without conducting new computer simulations. The drug-response profiles of four ion currents ($\delta_{CaL}$, $\delta_{Kr}$, $\delta_{NaL}$, $\delta_{Ks}$) are all that is needed to calculate the torsadogenic risk of the drug. The ion currents can be measured using standard patch clamp techniques and the risk metric can be calculated with pen and paper. This simplicity removes a technological hurdle to the adoption of computational assays in safety pharmacology.

### Identifying the axis of arrhythmia

All of the simulations in the present study were conducted in the absence of drugs. The simulations were only needed to identify the axis of arrhythmia and do not need to be repeated when applying the metric. The axis encapsulates the principal relationship between ion channel blockade and the onset of early after-depolarizations. It represents the most potent combination of pro-arrhythmic block that is theoretically possible for any drug, averaged across all cardiomyocytes. As such, it serves as an ideal yardstick for measuring the arrhythmogenic risk of real drugs.

The assumption that the boundary is linear is crucial for generalizing the findings across all cardiomyocytes. It allows the effect of a drug to be analyzed independently of the individual cardiomyocytes. So even though a nonlinear decision boundary might fit some cardiomyocytes better, it would not be helpful because the drug analysis would then be patient specific — which is not the aim of population safety pharmacology.

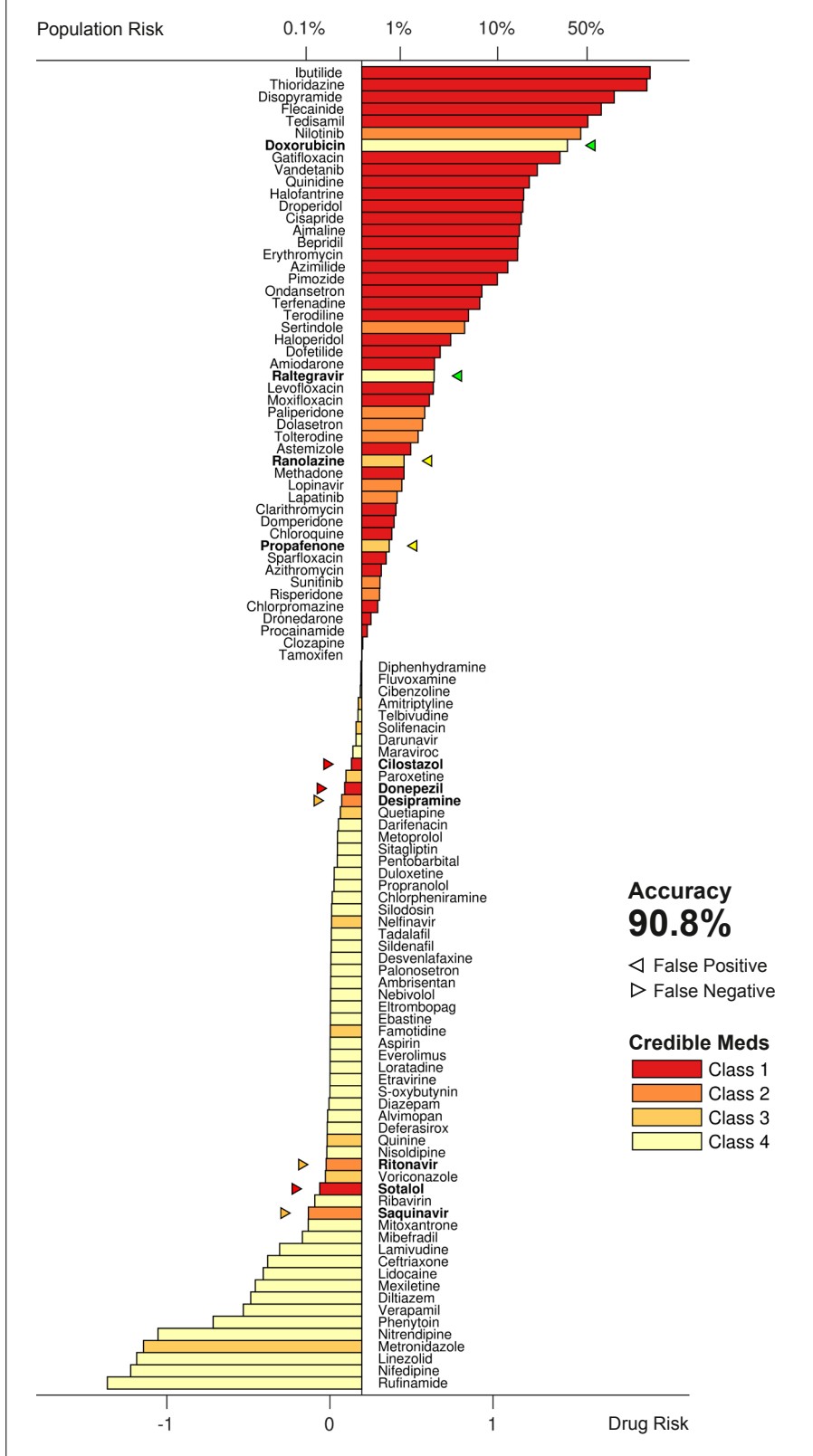

**Figure 7.** Torsadogenic risk for 109 drugs at 25× dose. Colors indicate the clinical risk labels from Credible Meds. The drugs are sorted by the score returned by our risk metric (lower axis). The proportion of the natural population that would be susceptible to the drug is shown on the upper axis. Drugs to the right of the scoring threshold ($\theta = 0.195$) were classified as unsafe and those to the left of it were classified as safe. Misclassified drugs are marked with a triangle and highlighted in bold. In this case, 90.8% of the drugs were correctly classified.

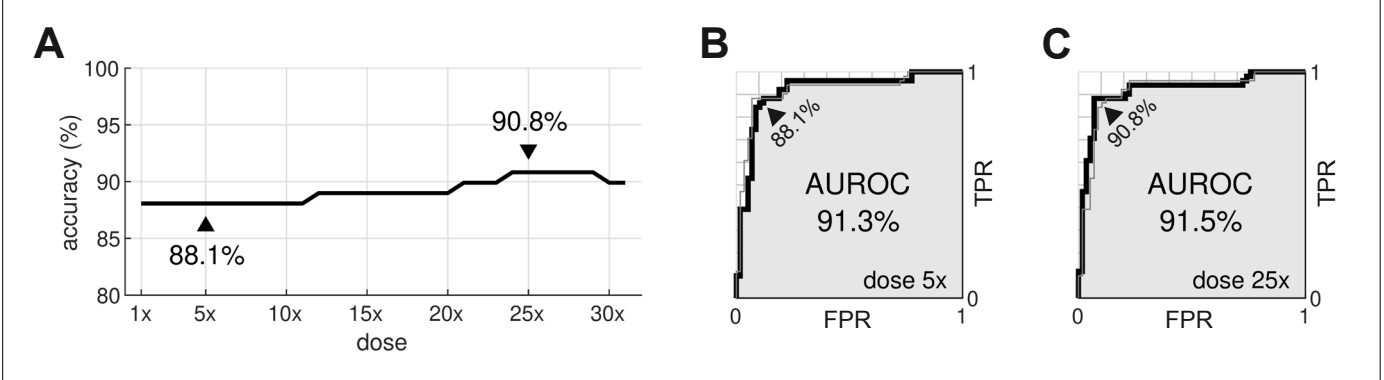

**Figure 8.** Optimal dosage. (**A**) Classification accuracy for drugs assessed at a range of dosages. (**B**) Receiver operating characteristic (ROC) curve for drugs at 5× dose. (**C**) ROC curve for drugs at 25× dose. AUROC is area under the ROC curve. TPR is true positive rate. FPR is false positive rate. The false negative rate is 1-TPR.

## Susceptibility to drug-induced arrhythmia

Our method can also predict the proportion of the population that would be susceptible to a drug without explicitly simulating it. The analysis is possible because drug blockade and phenotypic diversity both operate on the same properties of ion channels. The two biological processes are therefore mathematically interchangeable. As such, the distribution of the drugged population can be inferred from the drug-free population by shifting it according to the drug's risk score. The size of the susceptible population is a function of the risk score. So the torsadogenic risk can be reported either as a raw score or as a percentage of the population at risk.

## Comparison to conventional approaches

Conventional in silico safety assays are designed to apply the drug directly to simulated cardiomyocytes and then use biomarkers in the action potential to predict the torsadogenic risk. The biomarkers are typically optimized using a formal set of training drugs proposed by the comprehensive in vitro proarrhythmia assay (CiPA) initiative (*Colatsky et al., 2016*). See *Grandi et al., 2018*, for a review. The CiPA steering committee (*Li et al., 2019*) recommends that the electrophysiology be simulated using the CiPAORdv1.0 variant of the O'Hara-Rudy model which incorporates the kinetics of drugs binding with the hERG channel (*Li et al., 2017*). The recommended biomarker is qNet (*Dutta et al., 2017*) which measures the net charge of six major ion currents ($I_{Kr}, I_{CaL}, I_{NaL}, I_{to}, I_{Ks}, I_{K1}$). The clinical risks of the CiPA drugs (n=28) are labeled low, intermediate, or high. *Low* versus *intermediate-or-high* risk drugs were predicted with 84–95% accuracy using manual patch clamp techniques; or 93–100% accuracy using automated patch clamping (*Li et al., 2019*). Whereas *low-or-intermediate* versus *high* risk drugs were predicted to 92–100% accuracy using manual patch clamp; or 88–98% using automated patch clamp (*Li et al., 2019*). The measurement of the drug potencies is a source of considerable variability, which is exacerbated by the small number of test drugs (n=16). The prediction accuracy reported by *Li et al., 2019*, is higher than our method, but the results cannot be compared directly because the risk labels are stratified very differently.

*Passini et al., 2017*, used a larger dataset (n=62) to obtain 89% accuracy at predicting Class 1 versus Class 2–4 torsadogenic risk labels from Credible Meds. The cardiomyocytes were simulated using the baseline O'Hara-Rudy model without dynamic drug-binding kinetics. The accuracy is within the range of our results, but it too uses a slightly different risk stratification scheme. They scored the risk independently of dosage by averaging the number of early after-depolarizations, weighted by the concentration of the drug (*Passini et al., 2017*). We believe it is better to quantify the risk as a function of dosage, since even the most lethal drug is safe at zero dose.

The present study uses the same dataset (n=109) as *Llopis-Lorente et al., 2020*, who investigated a suite of biomarkers using a recalibrated O'Hara-Rudy model which also had revised gating kinetics for $I_{Na}$. Their best performing biomarker was qNet, which predicted Class 1–2 versus Class 3–4 torsadogenic risk with 92.7% accuracy. It exceeds our best result by approximately 2% on the same risk stratification scheme. *Llopis-Lorente et al., 2020*, increased the overall performance to 94.5% by

combining the best biomarkers into a decision tree. Interestingly, early after-depolarizations proved to be their worst biomarker, with only 78.9% accuracy, which is quite different to our findings. We suspect that their method under-estimated the predictive power of early after-depolarizations by only exploring normal physiological limits where such events are rare.

## Comparison to MICE models

MICE models (*Kramer et al., 2013*) are purely statistical. They use logistic regression to predict the torsadogenic risk of a drug directly from the half-maximal inhibitory concentrations ($IC_{50}$) for hERG, Cav1.2 and Nav1.5. *Kramer et al., 2013*, trained six candidate models on a dataset of 55 drugs. Their best model predicted the clinical risk labels of the drugs with 90.9% accuracy using only the difference between hERG and Cav1.2. Specifically,

$$\ln \frac{p}{1-p} = \beta_0 + \beta_1(C - H),$$

where $H = -\ln(IC_{50})$ for hERG and $C = -\ln(IC_{50})$ for Cav1.2. Their regression equation is strikingly similar to ours (*Equation 1*) except that MICE models use $\ln(IC_{50})$ for the predictor variables where we use $\ln(G)$. The distinction is that the $\beta$ coefficients in the MICE model have no biophysical interpretation, whereas in our model they are the basis of the axis of arrhythmia (*Equation 2*) and so represent the most potent combination of pro-arrhythmic channel blocks that is theoretically possible. Interestingly, the accuracy of the two models is nearly identical, albeit on different datasets.

## Comparison to Bnet

Bnet (*Mistry, 2018*) is a simple linear model that predicts the torsadogenic risk of a drug directly from the net blockade of inward and outward ion currents. Notably,

$$B_{\text{net}} = \sum_i^n R_r - \sum_j^m D_j$$

where $R_i$ and $D_j$ represent the percentage block of the repolarizing currents ($I_{\text{Kr}}, I_{\text{Ks}}, I_{\text{to}}$) and the depolarizing currents ($I_{\text{CaL}}, I_{\text{Na}}, I_{\text{NaL}}, I_{\text{K1}}$), respectively. Percentage block is akin to $(1 - \delta)$ in our model, but without the logarithmic transform. Bnet predicts the clinical risk labels of the CiPA validation drugs as accurately as the CiPAORdv1.0 model when adjusted for drug binding kinetics (*Li et al., 2019*; *Mistry, 2019*; *Han et al., 2019*). This has opened a debate between model complexity and biophysical realism in which proponents of biophysical models advocate their explanatory benefits (*Lancaster and Sobie, 2017*), whereas proponents of simple models advocate their predictive power without the computational expense (*Mistry et al., 2015*; *Mistry, 2017*).

## Conclusion

Our approach resolves the debate between model complexity and biophysical realism by combining both approaches into the same enterprise. Complex biophysical models were used to identify the relationship between ion channels and torsadogenic risk — as it is best understood by theory. Those findings were then reduced to a simpler linear model that can be applied to novel drugs without recapitulating the complex computer simulations. The reduced model retains a biophysical description of multi-channel drug block, but only as far as necessary to predict the likelihood of early after-depolarizations. It does not reproduce the action potential itself. Our approach thus represents a convergence of biophysical and simple models which retains the essential biophysics while discarding the unnecessary details. We believe the benefits of this approach will accelerate the adoption of computational assays in safety pharmacology and ultimately reduce the burden of animal testing.

## Limitations

The method was evaluated using a dataset of drugs that were drawn from multiple sources and diverse experimental conditions (*Llopis-Lorente et al., 2020*). It is known that such measurements differ prominently between laboratories and recording platforms (*Kramer et al., 2020*). Some drugs in the dataset combined measurements from disparate experiments while others had missing values. Of all the drugs in the dataset, only 17 had a complete set of $IC_{50}$ values for $I_{CaL}$, $I_{Kr}$, $I_{NaL}$, and $I_{Ks}$. The accuracy of the predictions is therefore limited by the quality of the drug potency measurements.

The accuracy of the axis of arrhythmia is likewise limited by the quality of the biophysical model from which it is derived. The present study only investigated one particular variant of the ORd model (*O'Hara et al., 2011*; *Krogh-Madsen et al., 2017*) paced at 1 Hz. Other models and pacing rates are likely to produce differing estimates of the axis.

## Methods

The action potentials of human endocardial ventricular cardiomyocytes were simulated using a variant of the ORD11 model (*O'Hara et al., 2011*) in which the maximal conductances of $G_{Ks}$, $G_{Kr}$, $G_{CaL}$, $G_{NaL}$, $P_{NaCa}$, and $P_{NaK}$ were re-scaled (*Table 1*) to better reproduce the clinical phenotypes of long QT syndrome (*Mann et al., 2016*; *Krogh-Madsen et al., 2017*). The source code for the ORD11 model (*O'Hara et al., 2011*) was adapted to run in the Brain Dynamics Toolbox (*Heitmann et al., 2018*; *Heitmann and Breakspear, 2022a*). The adapted source code is publicly available (*Heitmann, 2023*). The differential equations were integrated forward in time using the matlab ode15s solver with error tolerances AbsTol = 1e-3 and RelTol = 1e-6. The model was paced at 1 Hz with a stimulus of –70 mV and duration of 0.5 ms. All simulations were equilibrated for at least 1000 beats prior to analysis.

**Table 1.** The baseline scaling factors applied to the ORD11 model.

| Maximal conductance | Multiplier |
|---|---|
| $G_{Ks}$ | 8.09 |
| $G_{Kr}$ | 1.17 |
| $G_{CaL}$ ($P_{Ca}$) | 3.57 |
| $P_{NaCa}$ | 3.05 |
| $P_{NaK}$ | 1.91 |
| $G_{NaL}$ | 1.7 |

### Classification of ectopic phenotypes

Blocks of four successive beats were analyzed to accommodate alternans. Cells were classified as ectopic if any of those four beats contained an early after-depolarization — as defined by any secondary peak that rose above –50 mV and was separated from other peaks by at least 100 ms. Cells that did not exhibit early after-depolarizations were classified as benign.

### Parameter domain

We chose the domain of parameter space ($G \in e^{\pm 3}$) through trial and error. That domain is large enough to cover the ectopic region, but not so large as to be unduly influenced by biological extremes. The parameters span between 0.05 and 20 times their baseline value.

### Natural population density

Cell-to-cell variability was mimicked by scaling the conductances ($G_{CaL}, G_{Kr}, G_{NaL}, G_{Ks}$) by a random multiplier that was drawn from a log-normal distribution (*Sobie, 2009*). The parameters of the distribution ($\mu$=0.112, $\sigma$ = 0.472) were chosen to give the multipliers a mean of 1 and standard deviation of 0.5. The spread was based on our experience with previous simulations (*Sadrieh et al., 2013*; *Sadrieh et al., 2014*; *Ballouz et al., 2021*; *TeBay et al., 2022*). The ion channel species were assumed to vary independently.

By definition, the log-normal distribution maps onto the normal distribution under the logarithmic transform. The natural population is therefore represented in logarithmic parameter space by a symmetric multivariate normal distribution. Specifically,

$$f(w, x, y, z) = f(w) f(x) f(y) f(z),$$

where

$$f(x) = \frac{1}{\sqrt{2\pi\sigma^2}} \exp\left(-\frac{(x - \mu)^2}{2\sigma^2}\right) \tag{5}$$

is the univariate normal distribution.

**Table 2.** The drug-response dataset.

| Variable name | Description |
| --- | --- |
| DrugID | Unique identifier for each compound in the dataset |
| Compound | Name of the compound |
| Class | Clinical risk label where 1=*known* risk, 2=*possible* risk, 3=*conditional* risk, 4=*no evidence of risk* |
| EFTPC | Effective free therapeutic plasma concentration (nM) |
| Cmax | Concentration relative to therapeutic dose |
| Conc | Concentration of the dose (nM) |
| GKrScale | $\delta_{Kr}$ for the given dose (**Equation 7**) |
| GNaScale | $\delta_{Na}$ for the given dose (**Equation 7**) |
| GNaLScale | $\delta_{NaL}$ for the given dose (**Equation 7**) |
| GCaLScale | $\delta_{CaL}$ for the given dose (**Equation 7**) |
| GKsScale | $\delta_{Ks}$ for the given dose (**Equation 7**) |
| GK1Scale | $\delta_{K1}$ for the given dose (**Equation 7**) |
| GtoScale | $\delta_{to}$ for the given dose (**Equation 7**) |
| LogGKrScale | $\alpha_{Kr} = \ln(\delta_{Kr})$. |
|  | $\alpha_{Na} = \ln(\delta_{Na})$. |
| LogGNaLScale | $\alpha_{NaL} = \ln(\delta_{NaL})$. |
| LogGCaLScale | $\alpha_{CaL} = \ln(\delta_{CaL})$. |
| LogGKsScale | $\alpha_{Ks} = \ln(\delta_{Ks})$. |
| LogGK1Scale | $\alpha_{K1} = \ln(\delta_{K1})$. |
| LogGtoScale | $\alpha_{to} = \ln(\delta_{to})$. |
| Score | Risk score (**Equation 3**) |

The online version of this article includes the following source data for table 2:

**Source data 1.** The data in CSV format.

**Source data 2.** Dose-response plots for each drug in the dataset.

## Joint probability

Rotational symmetry allowed the multivariate distribution to be projected onto the axis of arrhythmia as a univariate distribution. The probability of a cardiomyocyte in the natural population being ectopic is

$$P = \int p(x) f(x) \, dx, \tag{6}$$

where $p(x)$ is the a priori probability of ectopy for that phenotype (**Equation 4**) and $f(x)$ is the proportion of cells in the population with that phenotype (**Equation 5**). Drugs serve to shift $f(x)$ along the axis of arrhythmia. The size of that shift is defined by the risk metric (**Equation 3**).

## Drug dataset

The drug-response curves were reconstructed using the Hill equation,

$$\delta = \frac{\mathrm{IC}_{50}^{h}}{\mathrm{IC}_{50}^{h} + [C]^{h}}, \tag{7}$$

where $\delta \in [0, 1]$ represents the fractional conductance of the ion channel. The concentration of the drug ($C$) was normalized to the effective free therapeutic plasma concentration (EFTPC). The EFTPCs and half-maximal inhibitory concentrations ($IC_{50}$) were taken from Supplementary Table S2 of *Llopis-Lorente et al., 2020*, which was curated from publicly available datasets and scientific publications. Where multiple values were encountered, *Llopis-Lorente et al., 2020*, used the median $IC_{50}$ and the worst-case (highest) EFTPC values. We assumed that ion channels with missing $IC_{50}$ values were not blocked by the drug ($\delta = 1$). We also fixed the Hill coefficients at $h = 1$ because (i) there is no evidence for co-operative drug binding in the literature, and thus no theoretical justification for using coefficients other than one; (ii) only 17 of the 109 drugs in the dataset had a complete set of Hill coefficients ($h_{CaL}, h_{Kr}, h_{NaL}, h_{Ks}$) anyway. The clinical risk labels for the drugs were transcribed from Table 1 of *Llopis-Lorente et al., 2020*.

### Dataset availability

The dataset containing the reconstructed drug-response curves is included in the source data for *Table 2*.

### Source code availability

The source code for the cell model is available from https://zenodo.org/records/7796721 under the GNU General Public License v3.0. The cell model requires version 2022 or later of the Brain Dynamics Toolbox (*Heitmann and Breakspear, 2022a*) which can be downloaded from https://zenodo.org/records/7070703 under the BSD 2-clause license.

## Acknowledgements

This study was supported by Australian NHMRC grants app1182032, app1182623, and NSW Health cardiovascular capacity building grant. SH was funded by The Medical Advances Without Animals Trust (MAWA) which aims to advance medical science and improve human health and therapeutic interventions without the use of animals or animal products. The Katana computing cluster on which the simulations were conducted is supported by Research Technology Services at UNSW Sydney.

## Additional information

### Funding

| Funder | Grant reference number | Author |
|---|---|---|
| National Health and Medical Research Council | app1182032 | Stewart Heitmann |
| National Health and Medical Research Council | app1142623 | Adam P Hill |
| NSW Health | Cardiovascular capacity building grant | Adam P Hill |
| Medical Advances Without Animals Trust | | Stewart Heitmann |

The funders had no role in study design, data collection and interpretation, or the decision to submit the work for publication.

### Author contributions

Stewart Heitmann, Conceptualization, Data curation, Software, Formal analysis, Funding acquisition, Validation, Investigation, Visualization, Methodology, Writing – original draft, Project administration; Jamie I Vandenberg, Adam P Hill, Conceptualization, Supervision, Funding acquisition, Investigation, Writing - review and editing

### Author ORCIDs

Stewart Heitmann ⓘ https://orcid.org/0000-0003-3351-9514
Jamie I Vandenberg ⓘ http://orcid.org/0000-0002-3859-3716

Reviewer #1 (Public Review): https://doi.org/10.7554/eLife.90027.3.sa1
Reviewer #2 (Public Review): https://doi.org/10.7554/eLife.90027.3.sa2
Author Response https://doi.org/10.7554/eLife.90027.3.sa3

## Additional files

### Supplementary files
• MDAR checklist

### Data availability

The drug dataset used in this study is included in the source data for Table 2. The source code for the cell model is available from https://zenodo.org/records/7796721 (*Heitmann, 2023*) under the GNU General Public License v3.0. The cell model requires version 2022 or later of the Brain Dynamics Toolbox (*Heitmann and Breakspear, 2022a*) which can be downloaded from https://zenodo.org/records/7070703 (*Heitmann and Breakspear, 2022b*) under the BSD 2-clause license.

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
