## [Editor Report · eLife assessment]

This **compelling** and novel mathematical method assesses drug pro-arrhythmic cardiotoxicity by examining the electrophysiology of untreated cardiac cells. It will be **valuable** for future drug safety design.

---

## [Referee Report · Reviewer #1 (Public Review)]

Summary:

Heitmann et al introduce a novel method for predicting the potential of drug candidates to cause Torsades de Pointes using simulations. Despite the fact that a multitude of such methods have been proposed in the past decade, this approach manages to provide novelty in a way that is potentially paradigm-shifting. The figures are beautiful and manage to convey difficult concepts intuitively.

Strengths:

(1) Novel combination of detailed mechanistic simulations with rigorous statistical modeling

(2) A method for predicting drug safety that can be used during drug development

(3) A clear explication of difficult concepts.

Weaknesses:

(1) In this reviewer's opinion, the most important scientific issue that can be addressed is the fact that when a drug blocks multiple channels, it is not only the IC50 but also the Hill coefficient that can differ. By the same token, two drugs that block the same channel may have identical IC50s but different Hill coefficients. This is important to consider since concentration-dependence is an important part of the results presented here. If the Hill coefficients were to be significantly different, the concentration-dependent curves shown in Figure 6 could look very different.

(2) The curved lines shown in Figure 6 can initially be difficult to comprehend, especially when all the previous presentations emphasized linearity. But a further issue is obscured in these plots, which is the fact that they show a two-dimensional projection of a 4-dimensional space. Some of the drugs might hit the channels that are not shown (INaL & IKs), whereas others will not. It is unclear, and unaddressed in the manuscript, how differences in the "hidden channels" will influence the shapes of these curves. An example, or at least some verbal description, could be very helpful.

---

## [Referee Report · Reviewer #2 (Public Review)]

Summary:

In the paper from Hartman, Vandenberg, and Hill entitled "assessing drug safety, by identifying the access of arrhythmia and cardio, myocytes, electro physiology", the authors, define a new metric, the axis of arrhythmia" that essentially describes the parameter space of ion channel conductance combinations, where early after depolarization can be observed.

Strengths:

There is an elegance to the way the authors have communicated the scoring system. The method is potentially useful because of its simplicity, accessibility, and ease of use. I do think it adds to the field for this reason - a number of existing methods are overly complex and unwieldy and not necessarily better than the simple parameter regime scan presented here.

Weaknesses:

The method described in the manuscript suffers from a number of weaknesses that plague current screening methods. Included in these are the data quality and selection used to inform the drug-blocking profile. It's well known that drug measurements vary widely, depending on the measurement conditions.

There doesn't seem to be any consideration of pacing frequency, which is an important consideration for arrhythmia triggers, resulting from repolarization abnormalities, but also depolarization abnormalities. Extremely high doses of drugs are used to assess the population risk. But does the method yield important information when realistic drug concentrations are used? In the discussion, the comparison to conventional approaches suggests that the presented method isn't necessarily better than conventional methods.

In conclusion, I have struggled to grasp the exceptional novelty of the new metric as presented, especially when considering that the badly needed future state must include a component of precision medicine.

---

## [Author Response]

The following is the authors’ response to the original reviews.

**Reviewer #1 (Public Review):**
Summary:Heitmann et al introduce a novel method for predicting the potential of drug candidates to cause Torsades de Pointes using simulations. Despite the fact that a multitude of such methods have been proposed in the past decade, this approach manages to provide novelty in a way that is potentially paradigm-shifting. The figures are beautiful and manage to convey difficult concepts intuitively.Strengths:(1) Novel combination of detailed mechanistic simulations with rigorous statistical modeling(2) A method for predicting drug safety that can be used during drug development (3) A clear explication of difficult concepts.Weaknesses:(1) In this reviewer's opinion, the most important scientific issue that can be addressed is the fact that when a drug blocks multiple channels, it is not only the IC50 but also the Hill coefficient that can differ. By the same token, two drugs that block the same channel may have identical IC50s but different Hill coefficients. This is important to consider since concentration-dependence is an important part of the results presented here. If the Hill coefficients were to be significantly different, the concentration- dependent curves shown in Figure 6 could look very different.

See our response below.

(2) The curved lines shown in Figure 6 can initially be difficult to comprehend, especially when all the previous presentations emphasized linearity. But a further issue is obscured in these plots, which is the fact that they show a two-dimensional projection of a 4dimensional space. Some of the drugs might hit the channels that are not shown (INaL & IKs), whereas others will not. It is unclear, and unaddressed in the manuscript, how differences in the "hidden channels" will influence the shapes of these curves. An example, or at least some verbal description, could be very helpful.

See our response below.

**Reviewer #1 (Recommendations For The Authors):**
The manuscript is generally well-written (with one important exception, see below). The manuscript can be improved with a few suggested modifications, ordered from most important to least important.(1) In this reviewer's opinion, the most important scientific issue that the authors need to address is the fact that when a drug blocks multiple channels, it is not only the IC50 but also the Hill coefficient that can differ. By the same token, two drugs that block the same channel may have identical IC50s but different Hill coefficients. This is important to consider since concentration-dependence is an important part of the results presented here.In a recent study (Varshneya et al, CPT PSP 2021 (PMID: 33205613)) they originally ran simulations with Hill coefficients of 1 for all the 4 drugs and 7 channels, then re-ran the simulations with differing Hill coefficients. The results were quantitatively quite different than what was originally obtained, even though the overall trends were identical. A look at the table provided in that paper's supplement shows that the estimated Hill coefficients range from 0.5 to 1.9, which is a pretty wide range.In this case, I don't think the authors should re-run the entire analysis. That would require entirely too much work and potentially detract from the elegant presentation of the manuscript in its current form. Although I haven't looked at the Llopis-Lorente dataset recently, I doubt that reliable Hill coefficients have been obtained for all 105 drugs. However, the Crumb et al dataset (PMID: 27060526) does provide this information for 30 drugs.Perhaps the authors could choose an example of two drugs that affect similar channels but with differences in the estimated Hill coefficients. Or even a carefully-designed hypothetical example could be of value. At the very least, Hill coefficients need to be mentioned as a limitation, but this would be stronger if it were coupled with at least some novel analyses.

We fixed the Hill coefficients to h=1 because there is no evidence for co-operative drug binding in the literature that would require coefficients other than one. There is also the practical matter that only 17 of the 109 drugs in the dataset have a complete set of Hill coefficients. We have revised the Methods (Drug datasets) to make these justifications explicit:

Lines 560-566: “… We also fixed the Hill coefficients at h = 1 because (i) there is no evidence for co-operative drug binding in the literature, and thus no theoretical justification for using coefficients other than one; (ii) only 17 of the 109 drugs in the dataset had a complete set of Hill coefficients (hCaL, hKr, hNaL, hKs) anyway. …”

Out of interest, we re-ran our analysis using only those n=17 drugs (Amiodarone,Amitriptyline, Bepridil, Chlorpromazine, Diltiazem, Dofetilide, Flecainide, Mibefradil,Moxifloxacin, Nilotinib, Ondansetron, Quinidine, Quinine, Ranolazine, Saquinavir, Terfenadine and Verapamil). When the Hill coefficients were fixed at h=1, the prediction accuracy was 88.2% irrespective of the dosage (Author response image 1). When we used the estimated (free) Hill coefficients, the prediction accuracy remained unchanged (88.2%) for all doses except the lowest (1x to 2x) where it dropped to 82.4%. We concluded that using the Hill coefficients from the dataset made little difference to the results.

(2) I initially had a hard time understanding the curved lines shown in Figure 6 when all the previous presentations emphasized linearity. After thinking for a while, I was able to get it, but there was a further issue that I still struggle with. That is the fact that the plots all show a two-dimensional projection of a 4-dimensional space. Some of the drugs might hit the channels that are not shown (INaL & IKs), whereas others will not. How will differences in the "hidden channels" influence the shapes of these curves? An example, or at least some verbal description, could be very helpful.

We omitted GKs and GNaL from Figure 6 because they added little to the story. Those “hidden” channels operate in the same manner as GKr and GNaL. They are shown in Supplementary Dataset S1. We have included more explicit references to the Supplementary in both the main text and the caption of Figure 6. We have also rewritten the section on ‘The effect of dosage on multi-channel block’ (lines 249-268) to better convey that the drug acts in four dimensions.

(3) I also struggled a bit with Figure 3 and the section "Drug risk metric." What made this confusing was the PQR notation on the figure and the equations represented as A and B. Can these be presented in a common notation, or can the relationship be defined?

We have replaced the PQR notation in Figure 3A with vector notation A and B to be consistent with the equations.

Also in Figure 3B, I was unclear about the units on the x-axis. Is each step (e.g. from 0 to 1) the same distance as a single log unit along the abscissa or ordinate in Figure 3A?

Yes it is. We have revised the caption for Figure 3B to explain it better.

(4) The manuscript manages to explain difficult concepts clearly, and it is generally wellwritten. The important exception, however, is that the manuscript contains far too many sentence fragments. These often occur when the authors explain a difficult concept, then follow up with something that is essentially "and this in addition" or "with the exception of this."Lines 220-223: "In comparison, Linezolid is an antibacterial agent that has no clinical evidence of Torsades (Class 4) even though it too blocks IKr. Albeit less than it blocks ICaL (Figure 5A, right)."Lines 242-245: "Conversely, Linezolid shifts the population 1.18 units away from the ectopic regime. So only 0.0095% of those who received Linezolid would be susceptible. A substantial drop from the baseline rate of 0.93%."There are several others that I didn't note, so the authors should perform a careful copy edit of the entire manuscript.

Thank you. We have remediated the fragmented sentences throughout.

**Reviewer #2 (Public Review):**
Summary:In the paper from Hartman, Vandenberg, and Hill entitled "assessing drug safety, by identifying the access of arrhythmia and cardio, myocytes, electro physiology", the authors, define a new metric, the axis of arrhythmia" that essentially describes the parameter space of ion channel conductance combinations, where early after depolarization can be observed.Strengths:There is an elegance to the way the authors have communicated the scoring system. The method is potentially useful because of its simplicity, accessibility, and ease of use. I do think it adds to the field for this reason - a number of existing methods are overly complex and unwieldy and not necessarily better than the simple parameter regime scan presented here.Weaknesses:The method described in the manuscript suffers from a number of weaknesses that plague current screening methods. Included in these are the data quality and selection used to inform the drug-blocking profile. It's well known that drug measurements vary widely, depending on the measurement conditions.

We agree and have added a new section to describe these limitations, as follows:

Lines 467-478: Limitations. The method was evaluated using a dataset of drugs that were drawn from multiple sources and diverse experimental conditions (LlopisLorente et al., 2020). It is known that such measurements differ prominently between laboratories and recording platforms (Kramer et al., 2020). Some drugs in the dataset combined measurements from disparate experiments while others had missing values. Of all the drugs in the dataset, only 17 had a complete set of IC50 values for ICaL, IKr, INaL and IKs. The accuracy of the predictions are therefore limited by the quality of the drug potency measurements.

There doesn't seem to be any consideration of pacing frequency, which is an important consideration for arrhythmia triggers, resulting from repolarization abnormalities, but also depolarization abnormalities.

It is true that we did not consider the effect of pacing frequency. We have included this in the limitations:

Lines 479-485: The accuracy of the axis of arrhythmia is likewise limited by the quality of the biophysical model from which it is derived. The present study only investigated one particular variant of the ORd model (O’Hara et al., 2011; KroghMadsen et al., 2017) paced at 1 Hz. Other models and pacing rates are likely to produce differing estimates of the axis.

Extremely high doses of drugs are used to assess the population risk. But does the method yield important information when realistic drug concentrations are used?

Yes it does. The drugs were assessed across a range of doses from 1x to 32x therapeutic dose (Figure 8A). The prediction accuracy at low doses is 88.1%.

In the discussion, the comparison to conventional approaches suggests that the presented method isn't necessarily better than conventional methods.

The comparison is not just about accuracy. Our method achieves the same results at greatly reduced computational cost without loss of biophysical interpretation. We emphasise this in the Conclusion:

Lines 446-465: Conclusion. Our approach resolves the debate between model complexity and biophysical realism by combining both approaches into the same enterprise. Complex biophysical models were used to identify the relationship between ion channels and torsadogenic risk — as it is best understood by theory. Those findings were then reduced to a simpler linear model that can be applied to novel drugs without recapitulating the complex computer simulations. The reduced model retains a bio-physical description of multi-channel drug block, but only as far as necessary to predict the likelihood of early after-depolarizations. It does not reproduce the action potential itself. Our approach thus represents a convergence of biophysical and simple models which retains the essential biophysics while discarding the unnecessary details. We believe the benefits of this approach will accelerate the adoption of computational assays in safety pharmacology and ultimately reduce the burden of animal testing.

In conclusion, I have struggled to grasp the exceptional novelty of the new metric as presented, especially when considering that the badly needed future state must include a component of precision medicine.

Safety pharmacology has a different aim to precision medicine. The former concerns the population whereas the latter concerns the individual. The novelty of our metric lies in reducing the complexity of multi-channel drug effects to a linear model that retains a biophysical interpretation.

**Reviewer #2 (Recommendations For The Authors):**
A large majority of drugs have more complex effects than a simple reduction and channel conductance. Some of these are included in the 109 drugs shown in Figure 7. An example is ranolazine, which is well known to have potent late sodium channel blocking effects - how are such effects included in the model as presented? I think at least suggesting how the approach can be expanded for broader applicability would be important to discuss.

Our method does consider the simultaneous effect of the drug on multiple ion channels, specifically the L-type calcium current (ICaL), the delayed rectifier potassium currents (IKr and IKs), and the late sodium current (INaL). In the case of ranolazine (class 3 risk), the dose-responses for all four ion channels, based on IC50s published in Llopis-Lorente et al. are given in Supplementary Dataset S1.

The response curves in Author response image 2 show that in this dataset, ranolazine blocks IKr and INaL almost equally - being only slightly less potent against IKr. There are two issues to consider here that potentially contribute to ranolazine being misclassified as pro-arrhythmic. First, the cell model is more sensitive to block of IKr than INaL. As a result, in the context of an equipotent drug, the prolonging effect of IKr block outweighs the balancing effect of INaL block, resulting in a pro-arrhythmic risk score. Second, the potency of IKr block in this dataset may be overestimated which in turn exaggerates the risk score. For example, measurements of ranolazine block of IKr from our own laboratory (Windley et al J Pharmacol Toxicol 87, 99–107, 2017) suggest that the IC50 of IKr is higher (35700 nM) than that reported in the LlopisLorente dataset (12000 nM). If this were taken into account, there would be less block of IKr relative to INaL, resulting in a safer risk score.

**Author response image 2. sa3fig2:**